# Combustion of Liquid Fuels in the Presence of CO$_2$ Hydrate Powder

Sergey Misyura [1,*], Vladimir Morozov [1], Igor Donskoy [2], Nikita Shlegel [3] and Vadim Dorokhov [3]

1   Kutateladze Institute of Thermophysics, Novosibirsk 630090, Russia; morozov.vova.88@mail.ru
2   Melentiev Energy Systems Institute SB RAS, Irkutsk 664033, Russia; donskoy.chem@mail.ru
3   Heat and Mass Transfer Laboratory, National Research Tomsk Polytechnic University, Tomsk 634050, Russia; nes6@tpu.ru (N.S.); dorohov45@yandex.ru (V.D.)
*   Correspondence: misura@itp.nsc.ru

**Abstract:** The process of combustion of a liquid fuel layer (diesel, kerosene, gasoline, separated petroleum, and oil) in the presence of CO$_2$ hydrate has been studied. These fuels are widely used in engineering, which explains the great interest in effective methods of extinguishing. Extinguishing liquid fuels is quite a complicated scientific and technical task. It is often necessary to deal with fire extinction during oil spills and at fuel burning in large containers outdoors and in warehouses. Recently, attention to new extinguishing methods has increased. Advances in technology of the production, storage, and transportation of inert gas hydrates enhance the opportunities of using CO$_2$ hydrate for extinguishing liquid fuels. Previous studies have shown a fairly high efficiency of CO$_2$ hydrate (compared to water spray) in the extinction of volumetric fires. To date, there are neither experimental data nor methods for determining the dissociation rate of CO$_2$ hydrate powder at the time of the gas hydrate fall on the burning layer of liquid fuel. The value of the dissociation rate is important to know in order to determine the temperatures of stable combustion and, accordingly, the mass of CO$_2$ hydrate required to extinguish the flame. For the first time, a method jointly accounting for both the combustion of liquid fuel and the dissociation rate of the falling powder of gas hydrate at a negative temperature is proposed. The combustion stability depends on many factors. This paper defines three characteristic modes of evaporation of a liquid fuel layer, depending on the prevalence of vapor diffusion or free gas convection. The influence of the diameter and height of the layer on the nature of fuel evaporation is investigated.

**Keywords:** combustion stability; carbon dioxide hydrate; gas hydrate dissociation; liquid fuel

## 1. Introduction

Social development and public safety depend on the effectiveness of the firefighting system [1]. To increase public safety, much attention is paid to forecasting fires at the community level [2], at the level of facilities, as well as to predicting risks in terms of material damage and human fatalities [3].

The features of fires and fire extinguishing depend on the place of spread and means of extinguishing. Methods of extinguishing fires in the forest area, in residential buildings, and in industrial and warehouse premises differ significantly. Industrial premises and warehouses with flammable materials present are ubiquitous in practice. Volumetric fire extinguishing in such premises is a complex scientific and technical task. The system of industrial fire-fighting methods is an important part of the entire production process. It is impossible to create a universal extinguishing agent for all types of fires and premises. Therefore, it is advisable to limit research to a narrower field of fire extinguishing agents. Fires in industrial premises are often associated with flammable liquids [4,5]. One of the causes of fires is associated with the leakage of liquid fuels during production, storage, and transportation [6]. The difficulty of extinguishing liquid fuels is associated with their

high fluidity, low heat of evaporation, and high heat release rate [7]. Another constraint in extinguishing such fires is associated with the rapid release and high concentration of harmful emissions, which require the creation of non-standard extinguishing methods. Gaseous combustion products of liquid fuels are dangerous for human respiratory organs. The extremely dangerous concentration of harmful impurities increases rapidly as a result of uncontrolled chemical reactions. Some of the widespread types of liquid fuels are gasoline, kerosene, and diesel fuel, which are used in the metallurgical sector, in the electric power industry, for transport, in the chemical industry, and in the aerospace industry [8–10].

Various fire extinguishing means and their applications are considered in [11–13]. These are the foaming agents that are most often used to extinguish liquid fuels [14–16]. However, it is difficult to quickly cover the entire surface of the burning fuel with foam over a large fire area, which reduces the extinguishing efficiency [17]. Fine droplets (water mist) are effectively used to extinguish massive fires. However, the efficiency of extinguishing combustible fuels with water mist is substantially lower than with foam as the mist is quickly removed from the combustion zone due to convection. Optimization of the parameters of fire extinguishing agents is considered in [18]. To increase the efficiency of extinguishing the oil fuels and refined fuels, surfactants reducing the size of gas bubbles in emulsions are used [19]. Iron pentacarbonyl and ferrocene are used as inhibitors of the chemical reaction during flame extinguishing [19,20]. The high toxicity of this extinguishing agent limits its use. Carbon dioxide is often used to dilute fire extinguishing agents when suppressing combustion [17,21].

Carbon dioxide may be used in the form of gas bubbles and dry ice, as well as in the form of $CO_2$ gas hydrate. Gas hydrates are solid crystalline compounds with "guest" gas and "host" water molecules that form crystal lattices with hydrogen bonds [22–24]. The crystalline equilibrium is provided by the van der Waals forces at certain pressure and temperatures. At disequilibrium, the gas hydrate breaks up into gas and ice. The process of gas hydrate dissociation at temperatures below the melting point of ice is much more complicated than that at positive temperatures. At subzero temperatures, porous ice forms, decreasing the dissociation rate tens to hundreds of times. The dissociation rate increases with an increase in the external heat flux [25]. During gas combustion, water droplets and water film appear on the surface of the gas hydrate powder layer, which affects the rate of dissociation [26]. The regularities of droplet evaporation are considered in [27].

In recent years, there has been a noticeable success in the development of technologies for storing and transporting gas hydrates at subzero temperatures, which enables a more efficient (less costly) delivery of inert gas hydrates to the flame region for the rapid suppression of combustion. The complicated development of these technologies is also due to the lack of elaborated methods for suppressing combustion, which requires additional experimental and theoretical studies. Combustion suppression with the use of $CO_2$ hydrate has to contend with a variety of related scientific tasks and key factors: convective heat and mass transfer; calculation of free convection, which strongly affects the evaporation rate of fuel; determination of criteria for combustion stability at inert gas hydrate powder entering the combustion region; and computation of the dissociation rate of gas hydrates at negative powder temperatures.

The practical application of carbon dioxide hydrate is considered in [28]. The method of seawater desalination due to $CO_2$ hydrate is described in [29]. The gas storage and transportation performance of $CO_2$ hydrate are discussed in [30–32]. The specifics of the production and use of carbonated solid products in the food industry are discussed in [33–35]. At dissociation, $CO_2$ hydrate breaks down into water and carbon dioxide. Water droplets, water vapor, and carbon dioxide are used for fire extinguishing. The heat of water evaporation, the heat of dissociation, and the melting of ice reduce the temperature in the flame zone. Incombustible gases released during the dissociation of gas hydrates prevent the entry of $O_2$ and pyrolysis products into the flame region [36,37]. Flame suppression due to the use of $CO_2$ hydrate is considered in [37]. Extinguishing flames with $CO_2$ hydrate powder has been poorly studied [38]. The critical mass of carbon dioxide hydrate necessary

to suppress flame burning is lower than that of ice. The amount of carbon dioxide released during flame extinguishing is less when using $CO_2$ hydrate compared with dry ice [37,38].

The characteristics of various combustible fuels answering the regulatory documents are given in [39–42]. The use of $CO_2$ for fire extinguishing is considered in [43]. One of the disadvantages of using carbon dioxide for flame extinguishing is a lack of oxygen, which is dangerous for human respiratory organs. However, high concentration of water vapor during the high-temperature dissociation of $CO_2$ hydrate significantly neutralizes the effects of carbon dioxide on humans [44].

It is important to note that the risk of fire and explosion in sections when using flammable liquids is much higher than when using solid combustible substances. Enhancing the efficiency of extinguishing liquid fuels requires new, more efficient methods of extinguishing. The use of carbon dioxide hydrate for this purpose will allow for creating more efficient extinguishing technologies based on the multicomponent composition. To date, this technology has not been given due attention, which has created a gap in knowledge and provided an impetus to the study.

Previously, experiments were carried out to extinguish the flame using $CO_2$ hydrate: (1) stopping the flame front, as well as (2) extinguishing a volumetric fire [45,46]. $CO_2$ hydrate powder demonstrated more effective extinguishing than water spray, snow, and ice. A particularly good result was observed for a volumetric fire extinguishing. Small drops of water did not fall into the lower part of the flame, evaporating in its upper part. Solid porous aggregates of gas hydrate fell into the bottom part of the tank until complete dissociation, providing flame suppression (lowering the temperature) over the entire height of the flame.

An analysis of the existing literature has shown that, to date, there are no methods for calculating flame suppression by $CO_2$ hydrate powder. The calculation is complicated by the necessity to simultaneously solve the related tasks: determining the rate of dissociation of $CO_2$ hydrate, determining the rate of formation of various components in the gas phase (air, carbon dioxide, fuel combustion products, and a high concentration of water vapor), and calculating the stability of combustion to these factors. The objectives of the work are as follows: (i) conducting experimental studies on extinguishing the flame from liquid fuels through the use of $CO_2$ hydrate and (ii) developing fundamentals for calculating the critical flame temperature (depending on the water vapor and carbon dioxide concentration) when extinguishing the flame through the use of $CO_2$ hydrate.

## 2. Experimental Technique

To increase the production rate of gas hydrate, finely crushed ice with a particle diameter of no more than 0.3 mm was used. Ice fragmentation allowed for achieving a high specific surface area. When the ice particles melted, a film of water appeared. The high gas pressure (above the equilibrium curve) inside the reactor ensured the synthesis of gas hydrate. The gas pressure was 5.5 MPa. The reactor temperature was 1 °C. The growth of the gas hydrate occurred within one day. Then, the gas hydrate was cooled to a temperature of about 150 K. At this temperature, the powder was taken out of the reactor, crushed, and put back into the reactor for synthesis. The resulting powder was sieved with sieves and the final average diameter of the gas hydrate particles was 0.3–0.4 mm. The $CO_2$ hydrate powder was stored in liquid nitrogen. In the experiments, a powder with an average diameter of 1–2 mm aggregates was used. The aggregation of fine powder into larger granules occurred spontaneously. Thus, $CO_2$ hydrate powder granules with a size of 1–2 mm were used in the experiments. The $CO_2$ hydrate had a cubic structure I. The formula of the elementary cell of $CO_2$ hydrate was $2D \cdot 6T \cdot 46H_2O$ or $2(5^{12}) + 6(5^{12}6^2)$ with consideration of the faces and crystal edges [22]. The initial concentration of the $CO_2$ hydrate was 20–21%. The porosity of the gas hydrate powder was 0.55–0.6. The porosity of the pressed tablets was 0.3. When the equilibrium was disturbed at negative temperatures, the gas hydrate broke up into ice and gas. At temperatures above the melting point of ice, carbon dioxide hydrate decomposed into water and gas.

The experiments employed AI-92 gasoline, TS-1 kerosene, diesel, oil, and separated oil, which are widely used in industry and are important in the study of fire extinguishing. The liquids had certificates of conformity and complied with standardized production conditions. The characteristics of the liquid fuels are given in Table A1 (Appendix A) [39–42]. The measured values of the fuel vapor front velocity at the beginning of combustion along the free surface of the fuel layer are given in Table A2.

Liquid fuel was poured into working sections (Figure 1). In the experiments, a working section with a diameter $D = 40$ mm and a height of the fuel layer $h = 3$ mm was used. The evaporation of liquid fuel during its combustion was investigated. The evaporation rate was measured using a weighting technique (Sartorius Secure scales with a discreteness of 0.0001 g). The working area with fuel was located on the scales (Figure 1). The evaporation rate was determined as $J = \Delta m / \Delta t$ ($m$ is the mass of fuel). The maximum error of the evaporation rate did not exceed 7%. The wall surface temperature was measured by thermocouples located at a distance of 1 mm from the wall with an error of 0.5–1 °C. The temperature of the free surface of the liquid layer before and after combustion was measured by the infrared camera NEC R500 with a resolution of $640 \times 512$ pixels and an error of 1–2 °C.

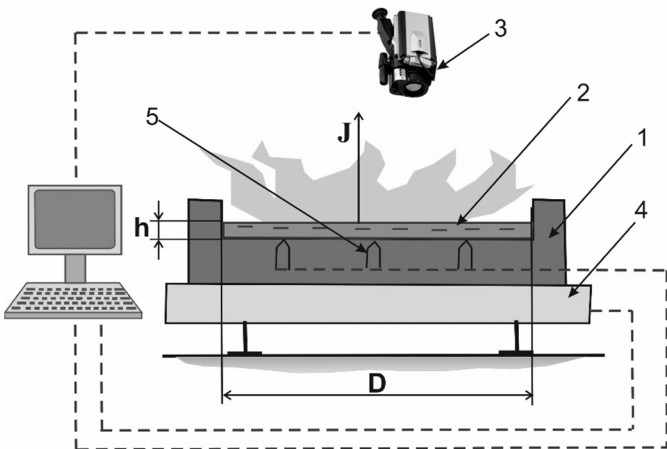

**Figure 1.** The scheme of the working unit for measuring the evaporation rate of liquid fuels: 1—cylindrical working section; 2—liquid fuel; 3—thermal imager; 4—electronic scales; 5—thermocouples for measuring the temperature of the wall surface.

## 3. Experimental Results and Discussion

### 3.1. Evaporation of the Liquid Fuel Layer during Combustion

To develop a technique for flame extinguishing using $CO_2$ hydrate, it is necessary to solve a number of interrelated tasks, associated with the following areas of research: (1) Heat and mass transfer and evaporation of liquid fuel during combustion, as well as in the presence of inert gas hydrate in the combustion region. (2) Determination of the dissociation rate of gas hydrate at negative powder temperatures. (3) Determination of the boundaries of flame stability when using $CO_2$ hydrate. These problems have not been solved to date, with each usually being investigated separately due to a large number of factors and uncertainty of boundary conditions. However, in order to optimize the extinguishing from an energy point of view (determining the minimum amount of $CO_2$ hydrate powder), it is expedient to solve these problems simultaneously.

For the correct solution of the thermal problem, it is important to determine the geometric parameters of the layer when it is necessary to take into account free convection, strongly affecting the fuel evaporation rate and the thermal balance in the combustion region. Therefore, the article examines the influence of the diameter of the liquid fuel layer on the kinetics of evaporation (the value of the exponent linking the layer diameter with the evaporation rate is determined).

Combustion kinetics depend on many parameters, including the concentration of fuel and inert gas. The task of determining the fuel concentration in the flame region, as well as in the presence of gas hydrate, is directly related to the determination of the fuel evaporation rate and the dissociation rate of $CO_2$ hydrate at negative powder temperatures. At that, negative temperatures are associated with the need to store the gas hydrate and transfer it to the combustion region without dissociation, until the powder enters the flame.

Flame extinguishing is aligned with determining the boundaries of stable combustion, which depend on the heat flux, determination of the combustion temperature, fuel evaporation rate, and dissociation rate of gas hydrate. In connection with the above, experimental studies and simplified calculation methods concern all these tasks.

To define the combustion of a liquid fuel layer, it is important to determine the effect of the layer height, layer diameter, and initial ignition temperature on the combustion kinetics. The layer height determines the time of the liquid heating. The layer diameter determines the type of dominant evaporation. For a small diameter of the layer, diffusion evaporation, vapor buoyancy (if the evaporating vapors have a lower density than air) [47], and Stefan flow prevail [48]. For large layer diameters, convective gas flow (free gas convection due to gas buoyancy) has a predominant effect on the evaporation rate of fuel vapors. The temperature of the fuel at which stable combustion begins depends on the type of fuel (heat of evaporation and heat of combustion).

Fuel mass change over time during combustion is shown in Figure 2. Time $t = 0$ s corresponds to the beginning of fuel combustion. The combustion of the fuel layer after its start quickly reaches a quasi-stationary state in the gas phase. The curves of all types of fuels have a linear character (for each investigated layer diameter). The curves for all fuel types have a linear character (at each studied layer diameter (19–80 mm)), which is associated with a constant equilibrium partial pressure of fuel vapor ($p_i$) during the entire combustion. The slope of the straight line generalizing the experimental points for gasoline is higher than for the other presented fuel. The minimal slope $m/m_0$ corresponds to oil.

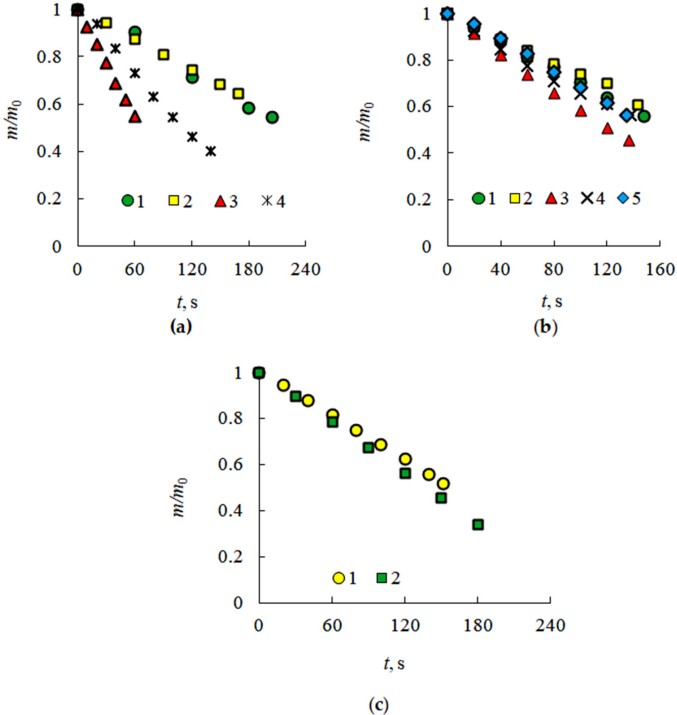

**Figure 2.** The change in the mass of liquid fuel over time during combustion ($m_0$—initial mass of fuel); (**a**)—layer diameter of 19 mm: 1—separated petroleum; 2—oil; 3—gasoline; 4—diesel; (**b**)—layer diameter of 40 mm: 1—separated petroleum; 2—oil; 3—gasoline; 4—diesel; 5—kerosene; (**c**)—layer diameter of 80 mm: 1—kerosene; 2—diesel.

Figure 3 shows the values of evaporation rate of various types of fuels during combustion. The maximal evaporation rate corresponds to gasoline, as the partial pressures of hydrocarbon vapors during gasoline evaporation are higher ($J_i \sim p_i$).

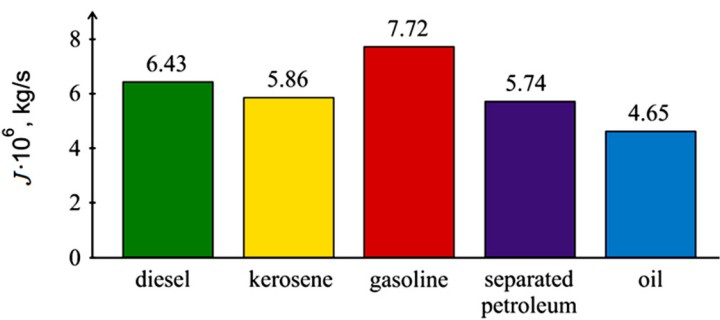

**Figure 3.** The rate of evaporation of fuels during combustion.

The evaporation rate of a thin layer of liquid in the absence of gas convection and steam buoyancy can be approximately described using Equation (1) [49],

$$J = \Delta m / \Delta t \approx -2D_l D \Delta \rho, \tag{1}$$

where the vapor density difference ($\Delta \rho = (\rho_s - \rho_{air})$) between the free liquid surface ($\rho_s$) and the external air ($\rho_{air}$), $D$ is the water vapor−air diffusion, and $D_l$ is the layer interface diameter. If the predominant influence on the rate of fuel evaporation is exerted by free gas convection, then $J$ is described by Equation (2) [50],

$$J = \beta_c F \Delta \rho, \tag{2}$$

where $\beta_c$ is the convective heat transfer coefficient and $F$ is the surface area of the layer ($F = \pi(D_l)^2/4$). If gas convection is neglected, the buoyancy of steam and the Stefan flow are proportional to the diffusion flow. In this case, $J \sim (D_l)^n$ ($n = 1$). If the predominant role in evaporation is played by convection in the gas phase, then $J \sim (D_l)^2$. Coefficient $\beta_c \sim (Re)^{0.5}$ ($Re = V_c R_l / \nu$), where $R_l$ is the radius of the layer and the buoyancy rate of the gas $V_c \sim (2g\beta(T_s - T_{air})R_l)^{0.5}$. Thus, with the growth of the layer diameter, the effect of convection on evaporation increases. Figure 4 shows experimental curves of the dependence of the evaporation rate of a liquid fuel layer on the layer diameter at a constant layer height of 2 mm.

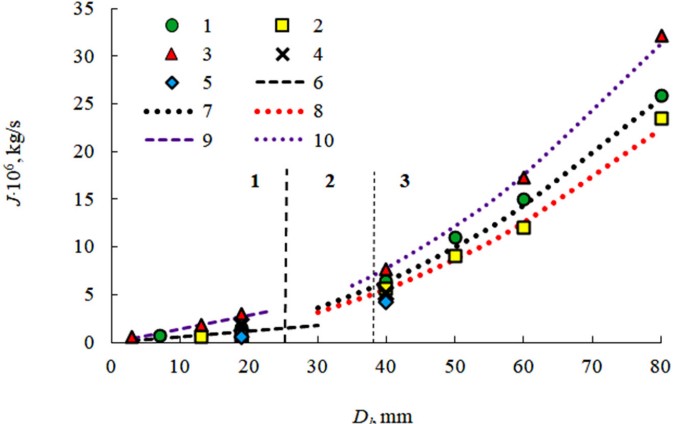

**Figure 4.** Dependence of the evaporation rate of the liquid fuel layer $J$ on the layer diameter during combustion (layer height of 2 mm; 1–5—experiment; 6–10—approximation curves: 1—diesel; 2—kerosene; 3—gasoline; 4—separated petroleum; 5—oil.

Mode 1 corresponds to diffusion evaporation ($J \sim (D_l)$ and mode 3 corresponds to convective evaporation $J \sim (D_l)^2$. For transition mode 2 ($J \sim (D_l)^n$), $n = 1–2$. As the diameter increases, the difference in $J$ between different types of fuels decreases.

Figure 5 shows the experimental values of the minimum temperature of liquid fuel at which stable spontaneous combustion is maintained. Different temperatures of liquids during combustion are important to take into account for modeling heat losses, which affect the temperature of the flame. Oil has the highest temperature of the liquid for stable spontaneous combustion. Thus, lowering the oil temperature (by the dissociation of the gas hydrate) below 200 °C will lead to the cessation of combustion. The temperature of the steady combustion of gasoline is quite low, so lowering the liquid temperature by evaporation is not effective. In this case, to stop combustion, it is necessary to lower the temperature in the burning area, as well as to reduce the concentration of combustible gases, due to their displacement by carbon dioxide during the dissociation of $CO_2$ hydrate.

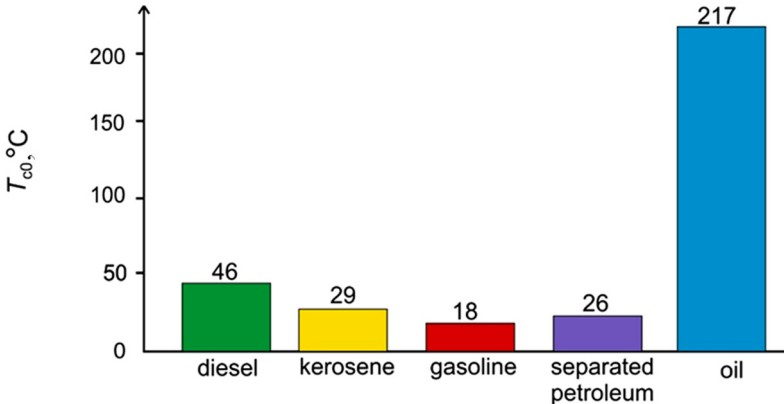

**Figure 5.** The minimum temperature of fuels for the realization of stable combustion.

Figure 6 shows the values of the wall temperature $T_w$ (the thermocouple is located in the center of the layer) at different heights of the liquid layer. The time $t = 0$ s corresponds to the beginning of combustion. The last points (maximum time) for all fuels correspond to the establishment of a quasi-constant wall temperature.

The temperature in the area of combustion of various fuels (in percentage terms) differs insignificantly compared with the temperatures of the liquid layer, which is associated with different heat of evaporation. The high heat of evaporation leads to a substantial decrease in the liquid temperature. The maximum evaporation rate and the maximum cooling of the liquid is realized during gasoline combustion. An increase in the liquid temperature during combustion by 50–55 °C is realized for both oil and gasoline (Figure 6a). As the layer height is 2 mm, the wall temperature ($T_w$) is close to that of the free surface of the layer ($T_s$). The maximum temperature increase is observed at kerosene layer combustion (over 90 °C). At a layer height of 12 mm (Figure 6b), a high transverse temperature gradient ($\Delta T = T_w - T_s$) is realized when burning a thick oil layer (over 100 °C). The time of the liquid layer heating (during combustion) from the initial temperature to the quasi-stationary one is shown in Figure 6c. With the increase in the fuel layer height from 2 to 12 mm, the time of the liquid heating to a quasi-stationary temperature increases 1.6–1.7 times. With an increase in the mass of liquid fuel (with an increase in $h$), the liquid takes longer to warm up to a quasi-stationary thermal regime. The heating rate of the fuel has a quasi-linear character depending on the layer height (the fuel mass).

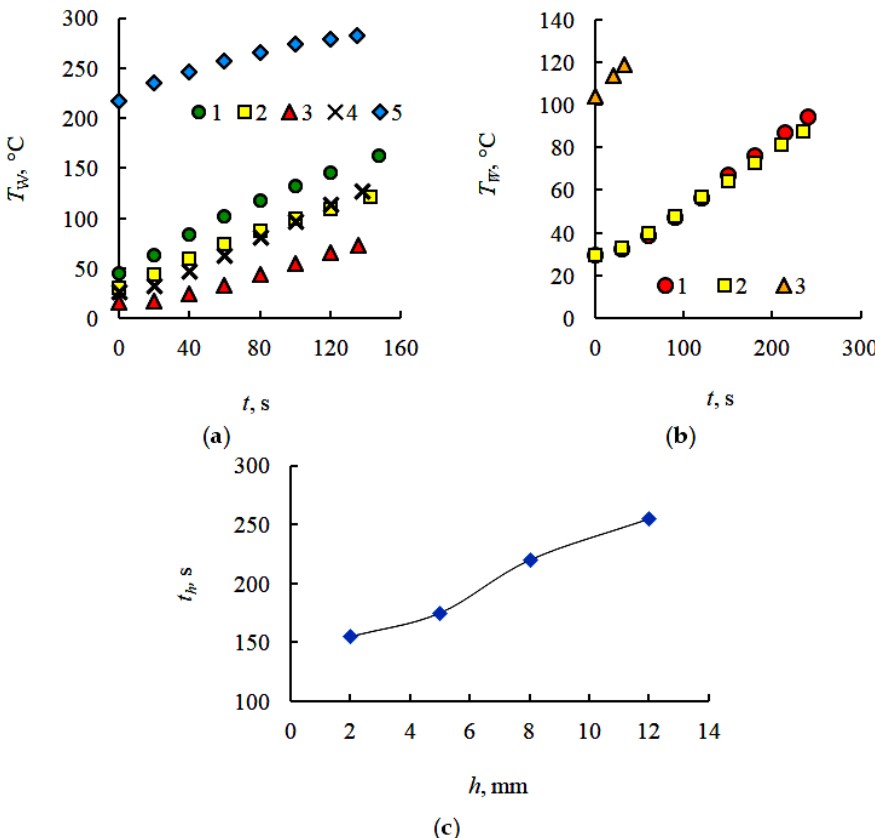

**(a)**

**(b)**

**(c)**

**Figure 6.** (**a**,**b**) Wall temperature change over time of the kerosene layer burning (40 mm layer diameter): (**a**) fuel layer height of 2 mm: 1—diesel; 2—kerosene; 3—gasoline; 4—separated petroleum; 5—oil; (**b**) fuel layer height of 12 mm: 1—diesel; 2—kerosene; 3—oil. (**c**) Dependence of the time of the kerosene layer heating on the fuel layer height (at a layer diameter of 40 mm).

Figures 7 and 8 show the thermal imaging measurements of the liquid fuel layer surface after combustion, as well as temperature profiles. The temperature $T_s$ on the layer surface is unevenly distributed. The higher temperature is observed near the side walls of the work area. After the combustion stops, the temperature of the liquid changes very slowly over time.

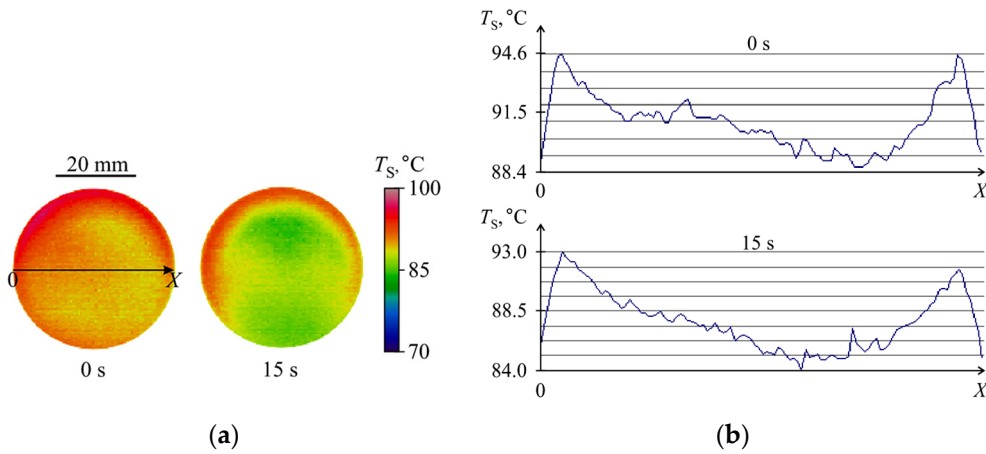

**(a)**

**(b)**

**Figure 7.** (**a**) Thermal image of the surface of the gasoline layer immediately after extinguishing the flame ($t = 0$ s corresponds to the time of extinguishing the flame). (**b**) Temperature profile of the gasoline layer surface along the lines $0X$ (from (**a**)).

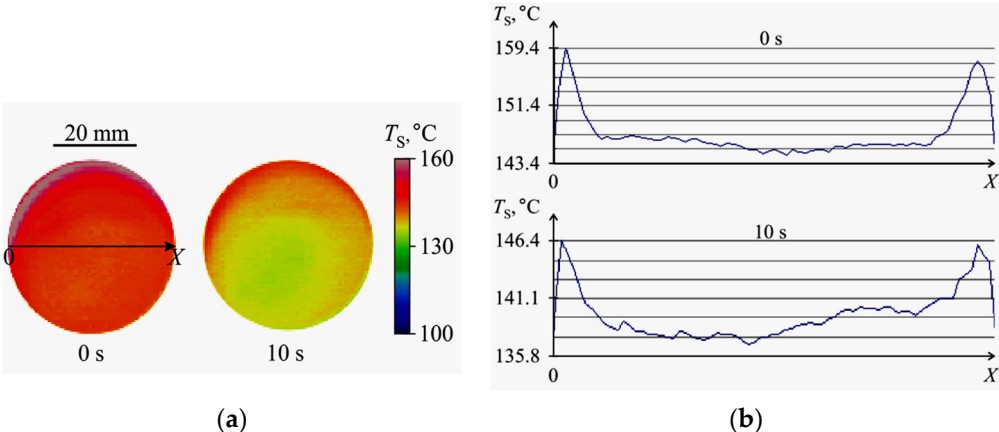

(a)　　　　　　　　　　　　　　　　　　(b)

**Figure 8.** (**a**) Thermal image of the kerosene layer surface immediately after extinguishing the flame. (**b**) Temperature profile of the kerosene layer surface along the 0*X* lines (from (**a**)).

Thus, immediately after combustion ceases, the combustion temperature $T_s$ is approximately equal to the layer surface temperature at the time of combustion. During combustion, the average surface temperature of the kerosene layer $T_s$ ($T_s$ = 147–149 °C) is much higher than the temperature of gasoline ($T_s$ = 90–91 °C). The heat loss from the combustion area (located at a distance from the free surface of the liquid layer) to the surface of the colder fuel will be noticeably higher for gasoline than for kerosene. Kerosene also has a higher specific heat of fuel combustion. However, the flame height is higher for gasoline (Figure 9a) than for kerosene (Figure 9b).

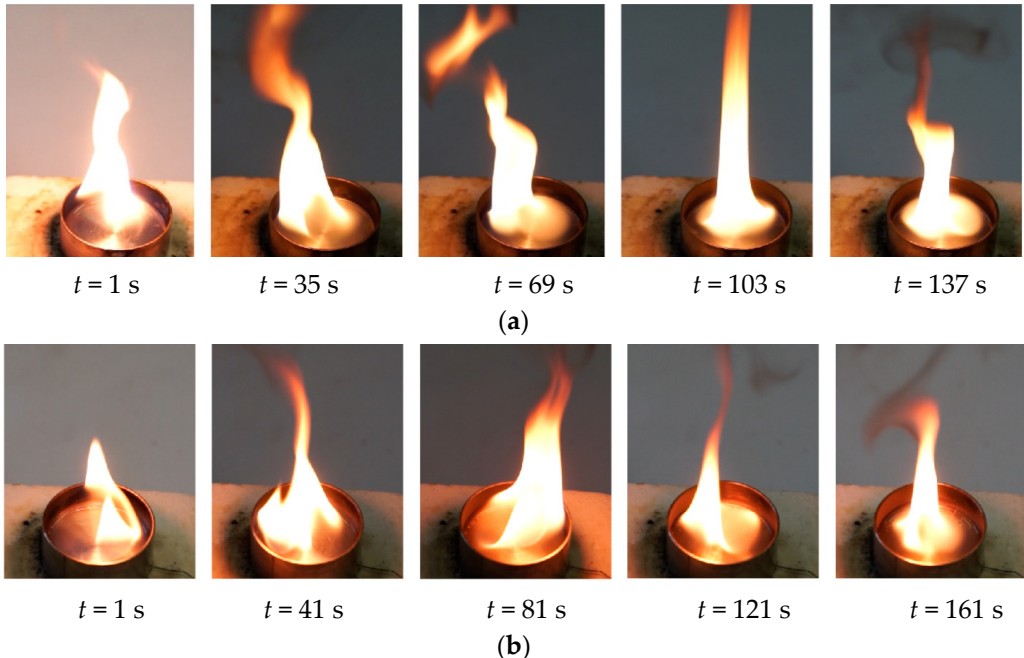

**Figure 9.** Combustion layer fuel (layer height of 2 mm, diameter of the working area of 40 mm): (**a**) gasoline and (**b**) kerosene.

The higher flame height for gasoline can be explained by the much higher transverse height of the diffusion layer due to the higher evaporation rate (Figure 3) and higher diffusion of fuel vapors. With an increase in the evaporation rate, the area of extremely low fuel vapor concentration (which is necessary for stable combustion) moves higher from the free surface of the fuel. Thus, it may be assumed that the distance y from the surface

of the fuel layer to the point of reaching the stoichiometric ratio (fuel/oxidizer) is higher for gasoline.

### 3.2. Evaporation of the Fuel Layer during Dissociation of the Carbon Dioxide Hydrate Tablet

After the beginning of combustion of a liquid layer (diesel, kerosene, and gasoline), a tablet of $CO_2$ hydrate was placed on the layer surface. To stop the combustion of diesel and kerosene, one pressed tablet was sufficient. The tablet diameter was 18–19 mm, the height was 4 mm, and the porosity was 30%. Even three tablets were not enough to extinguish the gasoline flame. The tablets were placed next to each other. After placing three tablets, the free surface of the gasoline layer (without tablets) occupied about 30% of the area of the entire surface of the layer. Figure 10 shows the experimental points for the change in the liquid layer mass $m/m_0$ (Figure 10a) and changes in the wall temperature $T_w$ (Figure 10b) before and after placing the gas hydrate tablets on the liquid surface (during fuel combustion).

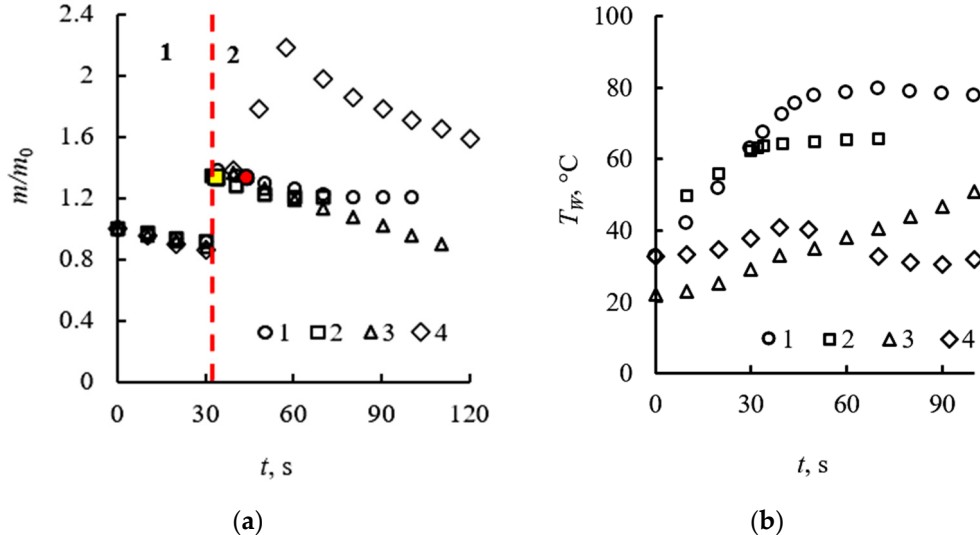

**Figure 10.** (**a**) Changes in the mass of the fuel layer before (area 1) and after (area 2) the placement of $CO_2$ hydrate tablets. (**b**) Changes in the wall temperature before and after the placement of gas hydrate tablets: 1—diesel; 2—kerosene; 3 and 4—gasoline (1, 2, and 3—one tablet; 4—three tablets, the fuel layer height is 2 mm and the layer diameter is 40 mm).

Curves 1 and 2 correspond to diesel and kerosene, respectively, and curves 3 and 4 correspond to gasoline: (3)—one tablet and (4)—three tablets. Color-filled symbols (dots) mean extinguishing the flame after placing the tablets. Unpainted dots indicate absence of flame extinguishing. The weight jump is due to the addition of tablets. For curves 1 and 2, there is a flame extinguishing. After extinguishing, the evaporation rate (1 and 2) decreases. The rate of fuel evaporation decreases due to a decrease in the heat flux to the liquid after extinguishing the flame. The mass of the tablet also decreases due to the gas hydrate dissociation. There is no quenching for curves 3 and 4.

It is characteristic that after placing the gasoline tablets (area 2), the slope of the mass over time was noticeably higher than without the tablets (area 1). With three tablets, about 70% of the fuel surface was occupied by $CO_2$ hydrate (the fuel evaporation surface has decreased three times). However, the slope of the mass curve increased markedly due to the gas hydrate dissociation (removal of carbon dioxide from the tablet). The removal of carbon dioxide with three tablets was obviously higher than with one tablet. For curves 1 and 2, the temperature $T_w$ stopped rising after the tablet was placed. For curve 3 (gasoline, one tablet), the temperature continued to rise even after the appearance of the tablet. Only after placing three tablets on the liquid layer (curve 3, gasoline), did $T_w$ decrease in the course of combustion. Thus, to extinguish gasoline (due to its low heat of evaporation, i.e.,

high volatility), it was necessary to cover almost the entire free surface of the fuel layer with gas hydrate.

Figure 11a,b shows experimental data on changes in the mass and temperature of the wall during the combustion of fuels with different layer diameters and different amounts of gas hydrate tablets. Symbols (dots) with a yellow color indicate the time of extinguishing the flame. With a four-fold increase in the surface area of the kerosene layer, three times more tablets were required to extinguish the flame (the mass of $CO_2$ hydrate increased three times).

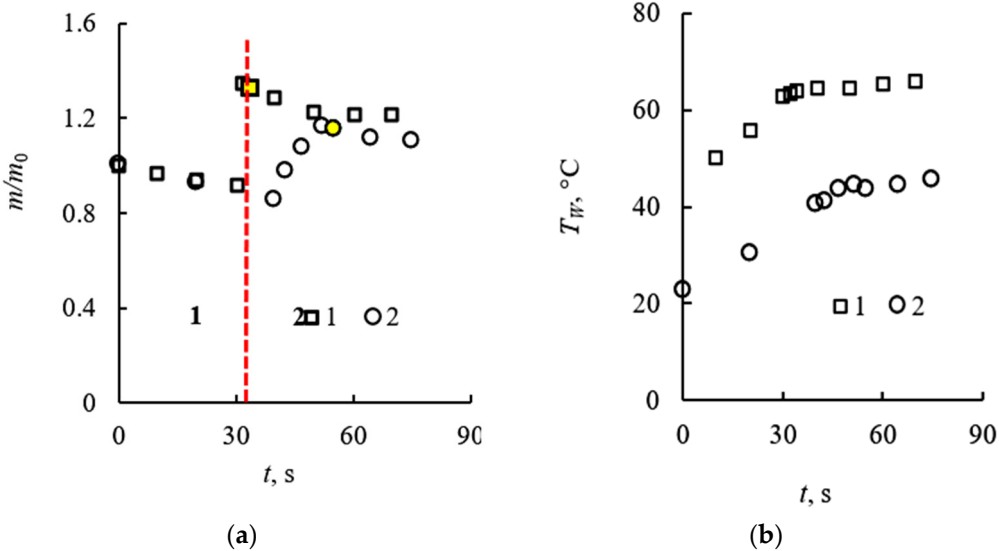

**Figure 11.** (**a**) Changes in the mass of the kerosene layer before (area 1) and after (area 2) the placement of $CO_2$ hydrate tablets. (**b**) Changes in the wall temperature before and after the placement of the gas hydrate tablets: 1—layer diameter of 40 mm, one tablet of gas hydrate; 2—80 mm, three tablets of $CO_2$ hydrate (fuel layer height of 2 mm).

## 4. Calculation of the Gas Hydrate Dissociation and Flame Extinguishing during Liquid Fuel Combustion

### 4.1. Dissociation of Carbon Dioxide Hydrate during Combustion

The difficulty in determining the flame extinguishing with gas hydrate lies in the joint solution of the problems of fuel combustion and dissociation of the falling powder of gas hydrate, as well as the determination of the boundaries of flame stability. Stability and temperature of combustion depend on the rate of hydrate dissociation (the rate of carbon dioxide release), which is unknown and can vary many times. To date, there are no reliable models for calculating the dissociation rate of gas hydrate at negative powder temperatures. The proposed method is the first stage of simplified modeling of joint combustion and dissociation for working out the main key factors and mechanisms. In the future, it is planned to expand the proposed methodology taking into account three-dimensional models and more complex combustion kinetics.

The dissociation rate of the gas hydrate $J = dm/dt$ at a temperature above the melting point of ice is determined by the pressure difference, in accordance with Equation (3) [51],

$$J = \frac{dm}{dt} = -c_0 \exp(-\frac{\Delta E}{RT})(P^{eq} - P^0)F \tag{3}$$

where $c_0$ is the internal kinetic constant, $\Delta E$ is the activation energy during the gas hydrate dissociation, $P^0$ is the gas pressure of the ambient medium, $P^{eq}$ is the equilibrium pressure of the gas hydrate, and $F$ is the surface area of the powder. At subzero temperatures, micro pores determining the dissociation rate are formed in the ice crust. Up to a temperature of $-45\,°C$, the pores are open and the dissociation rate is high. In the annealing temperature

window from $-45\,°C$ to $-5\,°C$, the pores are partially closed and the dissociation rate decreases by several orders of magnitude. At temperatures above $-4\,°C$, the fluidity of ice leads to the opening of pores and to a high rate of dissociation.

When a spherical particle of gas hydrate disintegrates, a two-layer system is formed: an outer shell of ice with pores and an inner core of gas hydrate. Upon gas hydrate dissociation, gas is released and removed to the outer space through the pores. Porosity must be taken into account for correct modeling of dissociation. It is convenient to consider the dissociation rate of a spherical particle in the form of Equation (4) [52,53],

$$\frac{dY}{dt} = -\frac{3k^R}{b\rho_H R_0} Y^{2/3}\left(P^{eq} + \frac{\gamma}{2} - \sqrt{\frac{\gamma^2}{4} + (P)^2 + \gamma P^{eq}}\right), \gamma = \frac{2R_0 k^R \mu R_g T}{k^F M_r}\left(Y^{\frac{1}{3}} - Y^{\frac{2}{3}}\right) \quad (4)$$

where the degree of transformation of a spherical gas hydrate particle $Y = m^H/m_0$ ($m^H$) is the current methane hydrate mass, $m_0$ is the initial mass of gas hydrate, $m^H = m^G/b$, $m^G$ is the mass of gas in the gas hydrate, $b$ is the initial mass concentration of methane, $R_0$ is the radius of the particle, $\rho_H$ is the density of the gas hydrate, $M_r$ is the molar mass of the gas hydrate, $R_g$ is the universal gas constant, $\mu$ is the dynamic viscosity of the gas, $T$ is the temperature, $k^R = k_0 exp(-E_a/RT)$ is the kinetic coefficient of the gas hydrate, $k^F$ is the permeability coefficient (it depends on the pore diameter $d_p$ and the pore density $\sigma_p$), $k^F = F_1(d_p)^2/32$, $F_1 = \sigma_p \pi (d_p)^2/4$ [54], and $F_1$ is the part of the particle surface that is occupied by pores. Modeling of the combustion of gas hydrates shows that a satisfactory generalization of experimental data on methane hydrate dissociation is realized through neglecting self-preservation, i.e., the filtration resistance of pores in the ice shell can be neglected [55]. In this case, the value of parameter $\gamma$ tends to zero. Then, Equation (4) will be simplified to become Equation (5).

$$\frac{dY}{dt} = -\frac{3k^R}{B\rho_H R_0} Y^{2/3}(P^{eq} - P^0) \quad (5)$$

Equation (5) is used to calculate the dissociation rate ($dY/dt$) of spherical particles of $CO_2$ hydrate when extinguishing a flame over liquid fuel (the powder consists of a given number of spherical particles defined by its porosity and particle diameter). The kinetic constant $k_0$ and the activation energy $E_a$ during the dissociation of carbon dioxide hydrate at negative temperatures were obtained in [56] ($k_0 = 0.0013\ kg/(m^2 Pa \cdot s)$, $\Delta E = 40.5\ kJ/mol$).

The efficiency of extinguishing the flame with carbon dioxide hydrate powder depends on the density of the $CO_2$ flow and the water vapor flow, which are determined by the gas hydrate dissociation rate. Heat transfer from the ambient gas to the surface of the gas hydrate particle is considered as convective heat transfer $q = \alpha(T_0 - T_s)$, where $T_0$ is the temperature of the ambient gas around the gas hydrate particle and $T_s$ is the surface temperature of the powder. Conductive heat flux inside a solid particle is $q = \lambda dT/dR$ ($\lambda$ is the thermal conductivity of ice and gas hydrate). On the surface of the particle ($R = R_0$), the boundary condition (Equation (6)) is satisfied ($\alpha$ is the heat transfer coefficient, $\lambda$ is the thermal conductivity (of gas hydrate/ice).

$$-\lambda \frac{\partial T}{\partial R}\bigg|_{R=R_0} = \alpha(T_0 - T_s) \quad (6)$$

As the particle is small (the particle diameter is 2 mm), the thickness of the water film can be neglected so as to consider only ice. Then, temperature $T_s$ is equal to the melting temperature of ice. As the powder is poured onto the area of the liquid burning from quite a height and the pouring process takes about 1 s, it is impossible to ensure uniform pouring. The scheme of the fall of the $CO_2$ hydrate powder and the extinguishing of the flame is shown in Figure 12a–c.

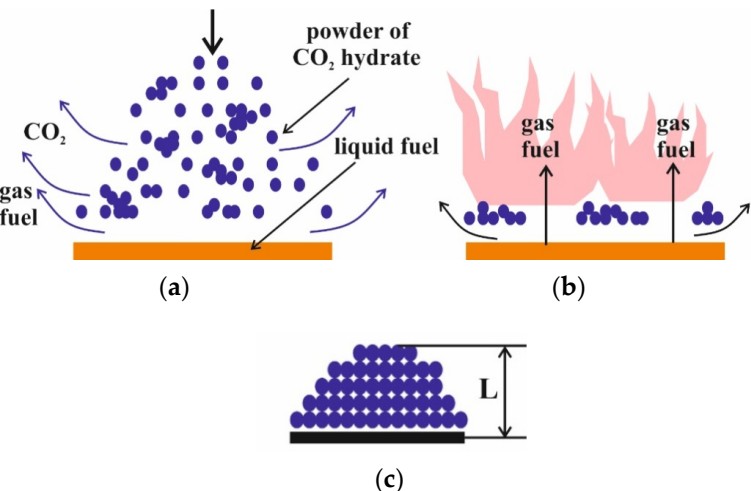

**Figure 12.** Scheme of flame extinguishing with $CO_2$ hydrate powder: (**a**,**b**) partial surface coverage; (**c**) full surface coverage.

When falling, the powder coalesces to create larger granules (Figure 12a,b), which are nonuniformly distributed over the volume. When the powder reaches the area of a high flame temperature, water vapor and $CO_2$ are produced. Part of the $CO_2$ and $H_2O$ is removed along the surface of the fuel. Most of the steam and carbon dioxide falls into the flame area, which leads to the squeeze-out of air (oxygen), a decrease in temperature, and extinguishing of the flame.

Extinguishing liquid and solid fuels with gas hydrates is fundamentally different. Unlike liquid fuel, solid fuel allows for extinguishing the flame without completely covering the fuel surface with gas hydrate [46]. This paper considers the extinguishing of flames of solid combustible materials (often located in residential and warehouse premises) with the help of $CO_2$ hydrate: cardboard, wood, and linoleum. If liquid fuel burning has stopped, but part of the surface of the fuel layer is not covered with gas hydrate, then due to the high volatility of the liquid (e.g., gasoline, alcohol) and high rate of fuel evaporation, burning may resume again.

### 4.2. Flame Extinguishing by $CO_2$ Hydrate Powder

The estimation of the stability limits of the diffusion flame of hydrocarbon vapors is based on the methodology used earlier in [57–60]. Calculations were carried out using the resources of the High-Temperature Circuit Multi-Access Research Center (project No. 13.CKP.21.0038) (Igor Donskoy).

Instead of the complete diffusion transfer equations with a chemical reaction in the region between the evaporating surface and the oxidizer flow, we considered an equivalent reactor of ideal mixing, in which the residence time $\tau$ is determined by the conditions of interfacial mass transfer. Despite its simplified nature, such a model allows for determining the stability limits of stationary diffusion flames; however, it requires careful choice of coefficients (first of all, heat transfer parameters). The gross reaction of complete oxidation of hydrocarbon fuel is described by Equation (7),

$$C_aH_b + \left(a + \frac{b}{4}\right)O_2 = aCO_2 + \frac{b}{2}H_2O \tag{7}$$

where $a = 8$ and $b = 18$ (in this case, the specific composition is insignificant and can be chosen with some degree of arbitrariness). Hydrocarbons, as a rule, exhibit more complex kinetic behavior (incomplete oxidation, multistage ignition, and soot formation). However, in the first approximation, complete oxidation corresponding to stationary diffusion combustion is considered. The interaction of fuel with water vapor and carbon dioxide is not taken into account. During the residence time $\tau$, the fuel reacts with the oxidizer according

to kinetic Equation (8), which reflects the difference in molar concentrations of fuel before and after the reactions,

$$n_F^{in} - n_F = \tau k_0 \exp\left(-\frac{E_a}{R_g T}\right) n_F^{\beta_1} n_{O_2}^{\beta_2} \tag{8}$$

where $n_F$ is the amount of fuel in the reaction mixture (the molar concentrations of components, mole/m³), index (*in*) refers to the initial composition, $k_0$ is the preexponent, $E_a$ is the activation energy, $R_g$ is the universal gas constant, and $\beta$ is the reaction order by the reagent. Equation (9) provides the concentrations of the components, which are determined in accordance with Equation (7).

$$n_{O_2} = n_{O_2}^{in} - \left(a + \tfrac{b}{4}\right)\left(n_F^{in} - n_F\right), n_{CO_2} = n_{CO_2}^{in} + a\left(n_F^{in} - n_F\right), n_{H_2O}$$
$$= n_{H_2O}^{in} + \tfrac{b}{4}\left(n_F^{in} - n_F\right) \tag{9}$$

Concentrations are related to temperature through the equation of state of an ideal gas according to Equation (10),

$$P_0 = R_g T \sum_j n_j \tag{10}$$

where $P_0$ is the pressure of the gas mixture equal to 1 bar. The values of kinetic coefficients are given in Table 1.

**Table 1.** Parameters used in the calculations.

| Parameter | Dimensions | Value, Source |
|:---:|:---:|:---:|
| $k_0$ | mol/m³/s | $1.42 \times 10^6$ [55] |
| $E_a$ | kJ/mol | 110 [55] |
| $\beta_1$ | - | 0.17 [55] |
| $\beta_2$ | - | 1.84 [55] |
| $L$ | m | 0.01 |
| $\tau$ | s | $10^{-3}$–$10^2$ |
| $T_0$ | K | 413 (experiment) |
| $R_g$ | kJ/mol/K | 8314 |
| $\lambda_g$ | W/m/K | 0.02 |
| $\kappa$ | 1/m | [61] |
| $\sigma$ | W/m/K⁴ | $5.67 \times 10^{-8}$ |

The thermal balance of an equivalent reactor (under isobaric conditions) can be written in the form of Equation (11),

$$\sum_j h_j^{in} n_j^{in} = \sum_j h_j^{out} n_j^{out} + \frac{\tau}{L}(q_{cond} + q_{rad}) \tag{11}$$

where $\tau$ is the average time of the gas presence in the reaction region; $L$ is the effective length of the combustion region, taking into account the volumetric heat losses (W·m⁻³K⁻¹); $h$ is the molar enthalpy (J·mole⁻¹); $h^{in}$ is the molar enthalpy at initial temperature; $h^{out}$ is the molar enthalpy at the temperature reaction; $n_j$ is the molar concentration of *j*-component (mole·m⁻³); and $q$ is the heat loss in the region of the flame. Thermodynamic data of substances are taken from the NASA database [61]. Heat losses may be written in accordance with Equation (12) [57],

$$q_{cond} = \frac{\lambda_g}{L}(T - T_0), q_{rad} = \varepsilon \kappa L \sigma \left(T^4 - T_0^4\right) \tag{12}$$

where $\lambda_g$ is the thermal conductivity of the gas, $L$ is the characteristic linear scale (the thickness of the diffusion layer of the combustion region), $\kappa$ is the absorption capacity of

the gas (determined by [62]), $\varepsilon$ is the degree of blackness of the surface (close to unity), $\sigma$ is the Stefan−Boltzmann constant, $T_0$ is the average temperature of the ambient medium (in calculations, this is the surface temperature of the liquid film on the surface of the particle).

The joint solution of Equations (6)–(9) is realized as follows. For the temperature range from the initial to maximum (adiabatic), kinetic Equation (6) is solved. The amounts of substances determined by Equations (7) and (9) are substituted in Equation (11), and as a result, the heat balance discrepancy is determined. In the temperature region, where the heat balance discrepancy changes sign, a search for clarifying a stationary temperature is performed. Under the given conditions of heat and mass transfer (i.e., for given values of characteristic time and linear scales), there may be from one to three stationary solutions [63]. The values of the parameters at which the number of solutions changes are critical and correspond to the limits of the stability of the diffusion flame. In the calculations, both boundaries correspond to extinction: at small residence times (due to low conversion of reagents) and at long residence times (due to increased heat loss). With a low fuel concentration, stable combustion becomes impossible, and the only solution is a low-temperature state for all of the values of the residence time. Dilution of fuel with water vapor and carbon dioxide caused by the addition of an extinguishing agent leads not only to a decrease in the reagent concentration, but also to an increase in the heat capacity of the reaction mixture [57]. Therefore, when assessing the boundaries of the existence of a diffusion flame, three options were considered: two limiting (dilution with water vapor and dilution with carbon dioxide) and one mixed (dilution with water vapor and carbon dioxide in equal proportions (50%/50%)).

The differences between different fuels are determined by their calorific value. The heat of combustion varies from 40 to 50 MJ/kg; this range covers most of the liquid fuels. As the characteristic residence time of fuel vapors in combustion depends on many factors, this parameter also varies in a wide range.

Figures 13 and 14 show curves of flame temperature change depending on the residence time ($\tau$) of the combustible gas in the combustion region. The curves stand for different volumetric degrees of fuel dilution $y$ (0–13 correspond to $y$ values).

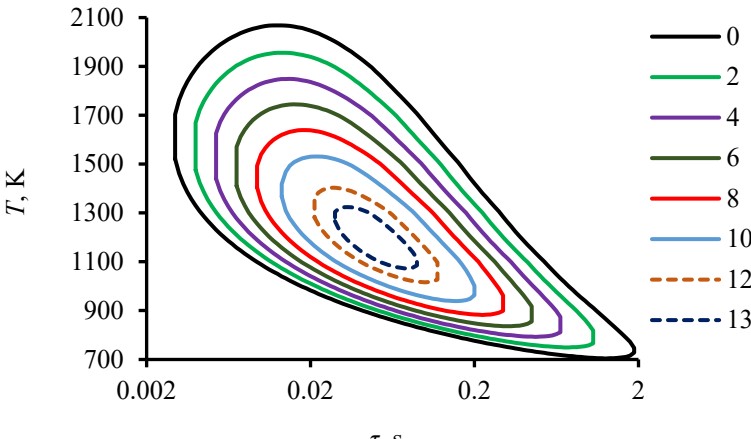

**Figure 13.** Changes in combustion temperature $T$ depending on time $\tau$ (the volume degree of dilution $y$ varies in the range of values 0–13; the combustible fuel is diluted only by $CO_2$ (100%)).

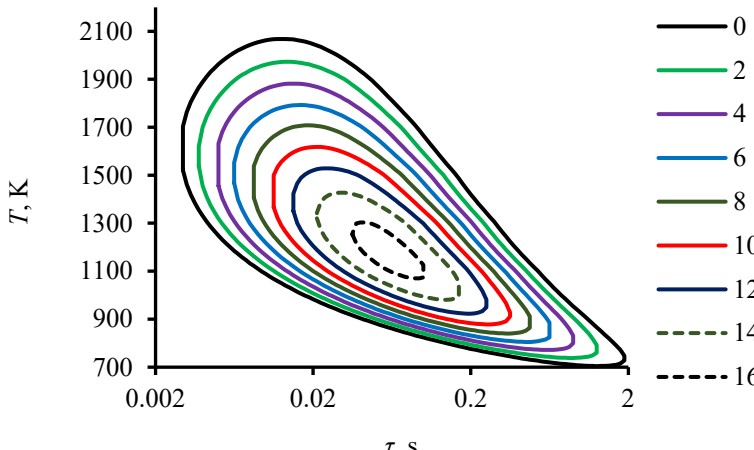

**Figure 14.** Changes in combustion temperature $T$ depending on time $\tau$ (the volume degree of dilution $y$ varies in the range of values 0–13; the combustible fuel is diluted with $CO_2$ (50%) and water vapor (50%)).

When carbon dioxide powder is poured into the burning fuel layer (Figure 12a), the gas hydrate decomposes into ice and $CO_2$. Under the influence of high temperature, ice melts, water evaporates, and carbon dioxide and water vapor enter the flame area along with fuel vapors, which leads to fuel dilution (lowering fuel concentrations in the flame area). The degree of fuel dilution with carbon dioxide $y = \frac{j_V}{j_F}\frac{\rho_F}{\rho_V}$, where $y$ is the volume dilution ratio, $j_V$ (kg·s$^{-1}$m$^{-2}$) is the flow of the $CO_2/H_2O$ mixture, $j_F$ (kg·s$^{-1}$m$^{-2}$) is the flow of fuel vapors, which is determined based on the experimental data, and $\rho_F$ and $\rho_V$ are the vapor densities of fuel and diluting gas, respectively.

The upper parts of the curves (straight lines) correspond to stable solutions, when stable combustion starts from an external heat source (for example, a spark or a gas burner). The lower parts of the curves (dotted lines) correspond to unstable solutions when the flame is extinguished in the absence of an external heat source. The maximum flame temperature approaches the maximum temperature in the absence of the release of carbon dioxide and water vapor from the gas hydrate. With the growth of $y$, the maximum temperature of stable combustion decreases. With the growth of $y$, the time interval $\tau$ for the implementation of sustainable solutions decreases sharply.

The mass consumption of carbon dioxide hydrate powder per unit surface of the liquid fuel layer ($m_\Sigma$, kg/m$^2$) is determined using Equation (13),

$$m_\Sigma = m_p y \frac{j_F}{j_p} \frac{M_V}{M_F} \tag{13}$$

where $m_p$ is the mass of one particle (granule) of $CO_2$ hydrate powder; $M_F$ and $M_V$ are the molecular masses of fuel vapor and diluting gas, respectively; $j_V = j_p\sigma$, where $j_p$ (kg/s) is the average mass velocity of one $CO_2$ hydrate particle decomposition (calculated in accordance with Equation (5)); and $\sigma$ is the surface density of particles, $N$ (number of particles)/$m^2$. In the calculation, the size of the aggregated $CO_2$ hydrate particles at the moment of entering the combustion region was taken at 1–2 mm. The mass of one $CO_2$ hydrate particle was $1.8 \times 10^{-6}$ kg. The average dissociation rate of one carbon dioxide hydrate particle was $4.5 \times 10^{-7}$ kg/s. The increase in the degree of dilution was achieved by increasing the number of particles. The mass consumption of powder ($m_\Sigma$) was equal to the product of the mass of one particle by the number of particles of carbon dioxide hydrate.

Figure 15a shows the calculated curves for the change in the maximum flame temperature of kerosene depending on the degree of fuel dilution $y$. Calculations were performed for three variants, when the dissociation rate during the $CO_2$ hydrate dissociation was determined only by carbon dioxide ($CO_2$ 100%) (curve 1), $H_2O$ 100% (curve 2), and $CO_2/H_2O$

(50%/50%, curve 3). For each value of $y$, a single curve and a single solution for the maximum flame temperature were realized (Figure 13). The curves in Figure 15 were obtained by solving Equation (11) for the enthalpies of the gas mixture in the flame region. In a wide range of $\tau$ for a given value of $y$, a curve of temperature change depending on $\tau$ was plotted (Figures 13 and 14). The maximum temperature was taken from the built curve. The termination of combustion corresponded to the critical value of $y_{cr}$ and the minimum temperature on the curve ($T_{cr}$). At a temperature $T_{max}$ below $T_{cr}$, as well as at a dilution degree $y$ above $y_{cr}$, combustion stopped. At $y > y_{cr}$, a stable solution (two stable roots of the equation) turned into a single solution (one root), in which the flame was extinguished regardless of the presence of an external heat source. For this case, there was not a closed curve, as in Figure 13, but a straight line, almost parallel to the abscissa axis. At a given $y$, regardless of the time value $\tau$, the temperature was constant due to the elimination of the oxidation reaction of fuel vapor.

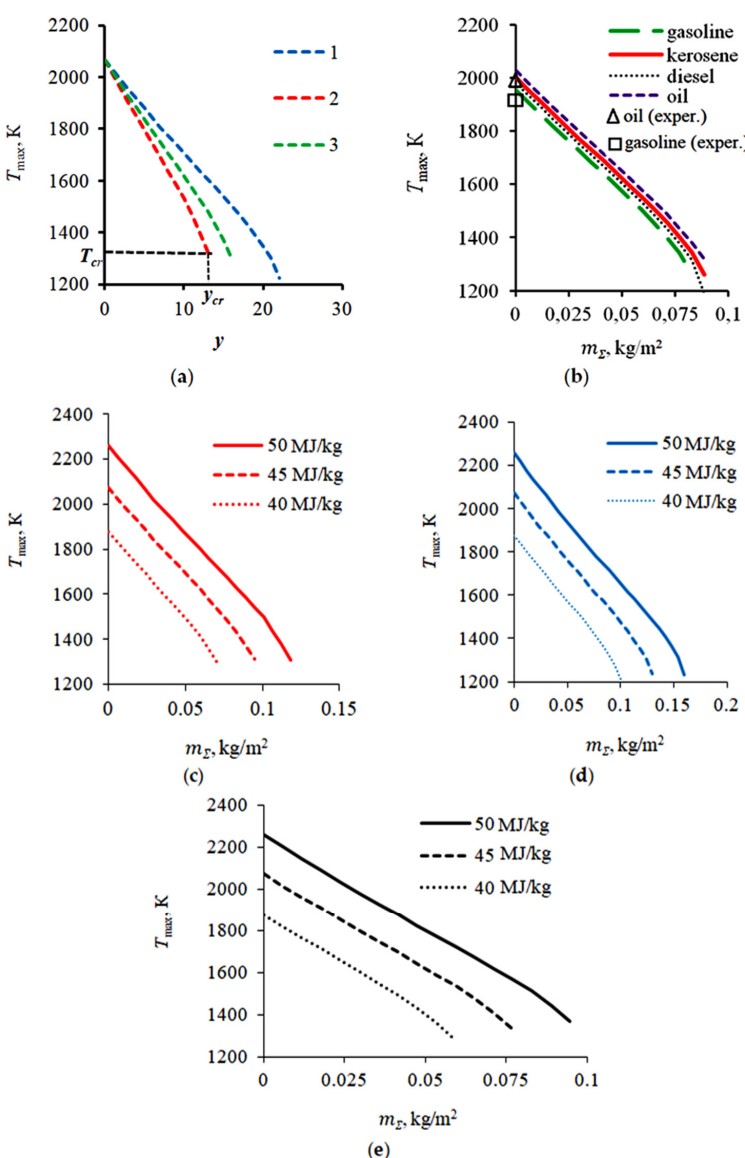

**Figure 15.** (**a**) The maximum flame temperature of kerosene depends on the degree of dilution of fuel $y$: 1—concentration of $H_2O$ C = 100%, concentration of $CO_2$ C = 0%; 2—$CO_2$ 100%; 3—$H_2O$ 50% and $C_2O$ 50%. (**b**) The maximum flame temperature depending on the mass consumption $m_\Sigma$; (**c**) $T_{max}$ with $m_\Sigma$ (50% $H_2O$/50% $CO_2$); (**d**) $T_{max}$ with the mass consumption (100% $H_2O$); (**e**) $T_{max}$ with $m_\Sigma$ (100% $CO_2$).

Obviously, with the same $y$ value (Figure 15a), the minimum flame temperature value corresponded to curve 2 ($H_2O$ 100%), as water vapor had a maximum heat capacity. The energy consumption was the greatest when heating the water vapor to the flame temperature. It is important to note that if most of the water did not fall into the combustion, the degree of dilution would be determined only by carbon dioxide. For example, when burning a pressed sphere of methane hydrate, most of the water film flowed down and water vapor practically did not lead to a decrease in the combustion temperature [64]. When extinguishing the flame with a $CO_2$ hydrate powder, almost all of the ice formed passed into water and most of the water vapor entered the combustion area. In this case, given that the mass concentration of carbon dioxide was 20–25%, water vapor had a predominant effect on extinguishing the flame.

Figure 15b illustrates the calculation of the drop in the maximum combustion temperature of fuels (gasoline, diesel, kerosene, and oil) depending on the mass consumption of $CO_2$ hydrate. In the absence of $CO_2$ hydrate, the maximum flame temperatures during the combustion of the gasoline and oil layer were measured to closely correspond to the calculated data. Depending on the ratio of carbon dioxide to water vapor in the area of combustion, the flame temperature changed significantly (Figure 15c). The higher one was the proportion of water vapor (40% of $CO_2$ and 60% of water vapor) the lower one was the combustion temperature. The displacement of oxygen by carbon dioxide from the combustion area led to combustion suppression. However, the displacement of water vapor, led to an increase in combustion temperature. Therefore, optimization of these concentrations is necessary.

The behavior of the maximum flame temperature during the combustion of conventional fuel (depending on the heat of combustion) is considered in Figure 15c–e. For the conventional fuel, the heat of combustion of 40, 45, and 50 MJ/kg was taken. In the combustion area, there was a mixture of gases (oxygen in stoichiometric ratio with fuel and air (nitrogen) to ensure oxygen/fuel stoichiometry).

Figure 15c shows the calculated curves for 50% carbon dioxide and 50% water (by mass fraction), formed during the gas hydrate dissociation, entering the flame region (during the $CO_2$ hydrate dissociation). Figure 15d corresponds to 100% steam entering the flame area. Figure 15e corresponds to 100% carbon dioxide entering the flame area. The effect of water vapor on the minimum amount of gas hydrate for extinguishing the flame was noticeably higher than that of the carbon dioxide. The maximum difference in $m_\Sigma$ in the range of 40–50 MJ/kg corresponded to carbon dioxide.

Recalculating the curves for dilution ratio $y$ into curves $m_\Sigma$ according to Equation (13) was not difficult. Figure 16a shows the calculated curves of heat flux losses due to conductive transport and radiation. With an increase in the amount of gas hydrate, the flame temperature decreased. As the flame temperature decreased, heat losses also decreased.

Heat losses due to the addition of carbon dioxide hydrate in relation to the released heat $q_F$ (formed as a result of fuel oxidation) are shown in Figure 16b. During $CO_2$ hydrate dissociation, heat losses $q_H = q_d + q_m + q_h + q_{ev}$, where $q_d$ is the heat of dissociation, $q_m$ is the heat of ice melting, $q_h$ is the heat of the water film heating, and $q_{ev}$ is the heat of evaporation. As the mass of the gas hydrate increased, $q_H/q_F$ increased linearly with the growth of $m_\Sigma$. The effect of the heat of the conventional fuel combustion (50% of $CO_2$/50% of $H_2O$) on $q_H$ is shown in Figure 16c.

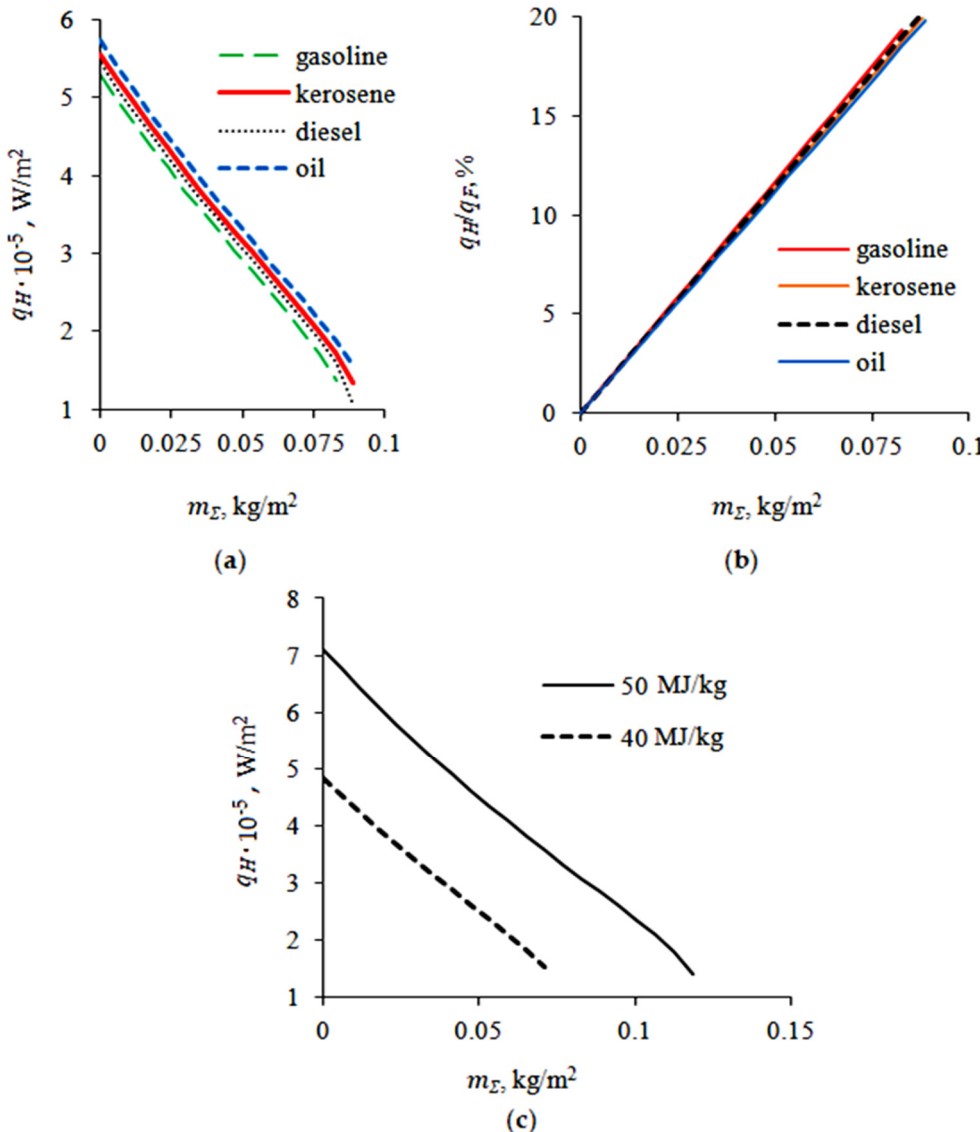

**Figure 16.** (**a**) Flame heat flux losses $q_H$ depending on the mass consumption of $CO_2$ hydrate powder $m_\Sigma$. (**b**) Heat flux losses $q_H/q_F$ ($q_H = q_d + q_m + q_h + q_{ev}$; $q_F$ is the fuel oxidation heat) associated with carbon dioxide hydrate depending on the mass consumption $m_\Sigma$. (**c**) Flame heat flux losses $q_H$ with $m_\Sigma$: (50% $CO_2$/50% of $H_2O$).

In the range of $m_\Sigma$ from 0 to 0.06 kg/m$^2$, the dependences are linear. When exceeding 0.06 kg/m$^2$, the linearity was slightly violated, which was associated with the kinetics of combustion (approaching the point of extinguishing the flame). Extinguishing the flame from the kerosene layer required a $CO_2$ hydrate mass that was about 6–7% larger compared with the gasoline layer. The higher value of $m_\Sigma$ for kerosene was associated with a higher combustion heat of kerosene (43.1 MJ/kg) compared with gasoline (42 MJ/kg). According to experimental data, 2–2.5 g of $CO_2$ hydrate is required to extinguish a layer of gasoline (kerosene) (the diameter of the layer is 65 mm). This powder mass consumption corresponds to 0.5 kg/m$^2$, which is 5–6 times higher than the calculated value of 0.085–0.9 kg/m$^2$. The excess of the experimental value over the calculated one is due to the fact that it is impossible to evenly and simultaneously scatter one layer of powder (Figure 12c) on a layer of burning fuel. Powder from a height of 100–200 mm was poured onto the surface of the fuel nonuniformly. In addition, the particles coalesced unevenly into larger aggregates (Figure 12a). If the mass of the powder corresponded to one layer of granules, then the coating nonuniformity would lead to a flow of fuel vapors from

certain areas of the surface that are not covered with powder (Figure 12b). In order to simultaneously cover the entire surface of the burning fuel, at least 5–6 layers are needed (Figure 12c). Thus, the effective $m_\Sigma$ ($m_{ef}$) for extinguishing the flame is $m_{ef} = k\, m_\Sigma$ ($k = 5$). Figure 17 shows data comparing the calculation with the experiment, taking into account the uniform pouring of the powder.

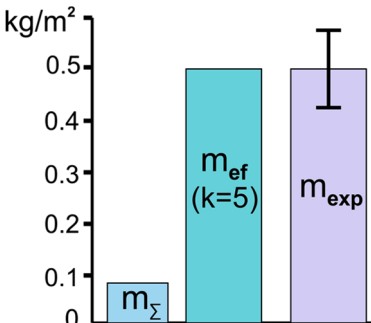

**Figure 17.** The mass consumption of $CO_2$ hydrate. $m_\Sigma$—calculation in the presence of one powder layer above the fuel surface; $m_{ef} = km_\Sigma$ ($k = 5$)—calculation for 5 powder layers; $m_{ef}$—experimental value of the minimum mass of the gas hydrate for extinguishing the flame.

Obviously, in the presence of wind, along with an increase in the height and area of pouring (the fire area), the coefficient $k$ can be greater than 5. However, the excess of $m_\Sigma$ of $CO_2$ hydrate powder over the calculated value was not related to the calculation method and did not take into account the unevenness of the pouring, the coalescence of granules, or the entrainment of granules by wind. To study these parameters, as well as to reduce the weight of the powder, further research is needed to optimize the technology of powder pouring.

Thus, the joint accounting of heat and mass transfer and fuel evaporation with free gas convection above the fuel layer, the fuel evaporation rate, and the gas hydrate dissociation rate, as well as the calculation of the critical rate of stable combustion, allows for applying a simplified methodology to determine the minimum mass of carbon dioxide hydrate required to suppress combustion. A thick layer of liquid fuel, as well as a volumetric fire, is the most difficult to extinguish [46]. As indicated in the Introduction, $CO_2$ hydrate powder shows a much greater efficiency compared with a water spray [46]. Large porous aggregates of $CO_2$ hydrate effectively and fall deep into the combustion area. However, the growth of the particle diameter, the diameter of the particle aggregate, and the height of the falling vapor layer lead to a decrease in the gas hydrate dissociation rate. The optimization of several key factors is necessary for the optimal selection of the diameter of aggregates, which is the subject of further research.

The use of pressed tablets and $CO_2$ hydrate powder are two extreme cases. Porous powder is best used to extinguish a flame when it is necessary to quickly cover the entire surface of the burning substance and ensure a high rate of carbon dioxide release (from the gas hydrate) for a split second. In this case, the flame height should be relatively low. If the distance to the place of fire is meters to tens of meters and the height of the flame is meters, then the powder with small particles can disintegrate before it reaches the surface of the burning substance. In this case, it is better to use large granules (tablets) of $CO_2$ hydrate, which will begin to disintegrate after falling on the burning surface, bypassing the intense and high flame. The size of the granules and their porosity will depend on the distance to the fire and the height of the flame. To determine the method of $CO_2$ hydrate delivery to the fire site, there is a need in additional experimental and theoretical studies for optimizing the geometric parameters of the gas hydrate and its mass.

## 5. Conclusions

For correct modeling of flame extinguishing using $CO_2$ hydrate, it is necessary to consider related processes of gas hydrate dissociation, liquid fuel evaporation, and gas fuel combustion during gas hydrate dissociation, as well as to determine the boundaries of combustion stability.

With the growth of the layer diameter, three characteristic modes of evaporation of the liquid fuel layer are realized, depending on the prevalence of vapor diffusion or free gas convection. The evaporation mode is important to take into account for the correct modeling of the fuel evaporation rate and combustion temperature. An increase in the rate of fuel evaporation due to gas free convection leads to a higher mass flow of $CO_2$ hydrate for fuel extinguishing.

The evaporation rate of the gasoline layer during combustion is almost 40% higher than kerosene. The evaporation rate is constant over the burning time. The surface temperature of kerosene during combustion is much higher than that of gasoline. The steady burning of gasoline without an external heat source requires a higher heating of kerosene compared with gasoline. The paper presents the calculated curves of the combustion stability of liquid fuel (gasoline/kerosene) in a wide range of carbon dioxide and water vapor flow rate, which are formed during the dissociation of gas hydrate. With increasing the flow rate of carbon dioxide and water vapor ($y$), the maximum temperature of stable combustion decreases. With the growth of parameter $y$, time interval $\tau$ (the residence time of fuel and oxidizer vapors in the flame region) for the implementation of stable solutions decreases sharply.

The calculation method determining the minimum mass consumption of $CO_2$ hydrate powder for extinguishing liquid fuel (gasoline/kerosene) is presented. The importance of instantaneous blockage of the entire surface of the burning fuel layer to prevent the access of combustible vapors to combustion area is shown. Unlike liquid fuel, solid fuel allows for extinguishing the flame without completely covering the free surface of the fuel layer with gas hydrate.

Water vapor has a stronger effect on extinguishing the flame (compared with carbon dioxide). As the flame temperature decreases, heat losses also decrease. Extinguishing the flame from the kerosene layer requires about 6–7% larger mass of $CO_2$ hydrate than extinguishing the gasoline layer, which is associated with a higher value for the combustion heat of kerosene compared with gasoline.

After placing $CO_2$ hydrate tablets during fuel combustion, the mass slope over time was noticeably higher than without tablets. With three tablets (about 70% of the fuel surface was occupied by $CO_2$ hydrate), the evaporation surface of the fuel decreased three times. The slope of the mass curve increased markedly due to the dissociation of the gas hydrate during the removal of carbon dioxide from the tablets.

To extinguish a volumetric fire with a high flame height, as well as at a great distance to the fire site (meters-tens of meters), it is efficient to use pressed tablets or granules. In this case, the gas hydrate tablet will not have time to disintegrate until it falls into the lower region of the flame. In some cases of local combustion and low flame height (less than 0.5–1 m), it is possible to use a highly porous powder with $CO_2$ hydrate particles of small diameter (about 1–2 mm or less) to extinguish the flame.

**Author Contributions:** Methodology, S.M., I.D., N.S. and V.D.; investigation, S.M. and V.M.; writing—review and editing, S.M.; visualization, S.M. and V.M. All authors have read and agreed to the published version of the manuscript.

**Funding:** The study was supported by a grant from the Ministry of Science and Higher Education of Russia, Agreement of 29 September 2020 Number 075-15-2020-806 (Contract Number 13.1902.21.0014).

**Institutional Review Board Statement:** Not applicable.

**Informed Consent Statement:** Not applicable.

**Data Availability Statement:** Not applicable.

**Conflicts of Interest:** The authors declare no conflict of interest.

## Appendix A

**Table A1.** Characteristics of flammable liquids in compliance with [39–42].

| | | |
|---|---|---|
| AI-92 gasoline | Octane number | 91 |
| | Lead content, g/dm$^3$ | 0.01 |
| | Manganese content, mg/dm$^3$ | 18 |
| | Oxidation stability of gasoline, min | 360 |
| | Existent gum content, mg/100 cm$^3$ | 5 |
| | Mass fraction of sulfur, % | 0.05 |
| TS-1 kerosene | Density at 20 °C, g/cm$^3$ | 0.780 |
| | Kinematic viscosity, mm$^2$/s at 20 °C | 1.3 |
| | Lower heating value, kJ/kg | 43,120 |
| | Mass fraction of total sulfur, % | 0.2 |
| Diesel | Cetane number | 45 |
| | Kinematic viscosity, mm$^2$/s at 20 °C | 4 |
| | Ash content, % | 0.01 |
| | Concentration of actual resins, mg per 100 cm$^3$ of fuel | 30 |
| Oil (Lukoil Genesis Armortech 5W30) | Density at 15 °C, kg/m$^3$ | 844.8 |
| | Kinematic viscosity at 100 °C, mm$^2$/s | 9.7 |
| | Viscosity index | 173 |
| | Sulphate ash content, % | 1.05 |
| | Flash point in an open crucible, °C | 223 |
| Oil separated | Mass fraction of water, wt.% | Absent |
| | Density at 15 °C, kg/m$^3$ | 797.5 |
| | Density at 20 °C, kg/m$^3$ | 792.5 |
| | Kinematic viscosity at 15 °C, mm$^2$/s | 4.391 |
| | Kinematic viscosity at 20 °C, mm$^2$/s | 1.741 |

**Table A2.** Propagation velocities of the liquid vapor combustion fronts.

| Fuel | Velocity, m/s |
|---|---|
| AI-92 gasoline | 1.4 |
| TS-1 kerosene | 0.93 |
| Separated oil | 1.3 |
| oil | 0.8 |
| Diesel | 0.1 |

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
