# Peer review of "Combustion of Liquid Fuels in the Presence of CO2 Hydrate Powder"

_fire, doi:10.3390/fire6080318_

Round 1
Reviewer 1 Report
Review
to the manuscript “Combustion of liquid fuels in the presence of CO2 hydrate powder”
The manuscript explores combustion of the liquid fuel layer of diesel, kerosene, gasoline, separated petroleum and oil in the presence of CO2 hydrate. The method for joint accounting of both the liquid fuel combustion and the dissociation rate of the falling powder of gas hydrate is proposed. It is shown that the combustion stability depends on a large number of factors: combustion kinetics, the gas hydrate dissociation kinetics and the convective heat. The gravimetric method for measuring the fuel evaporation rate was used. Temperature measurements on the free layer surface by thermal imaging technique were performed. The manuscript defines 3 evaporation modes of liquid fuels, depending on the prevalence of vapor diffusion or free gas convection. The influence of the layer diameter and the layer height on the fuel evaporation nature is investigated.
The received material is original and is important for researchers. Despite the fact that today there are no technologies for extinguishing fires using inert gases with gas hydrates, these studies are important from the point of view of the prospects for the development of new methods, as well as from a fundamental point of view.
To date, the problems of modeling the dissociation of the falling powder of carbon dioxide hydrate into the combustion region have not been solved. A joint solution of gas hydrate dissociations at negative temperatures, as well as a description of heat and mass transfer and fuel combustion is required. The material of the article successfully helps to formulate and approximately solve these questions.
The literature review was carried out carefully. The history of the problem is shown. The calculation methods are described in detail. The article is written clearly, indicating important physical factors.
There are a number of issues that need to be clarified:
1. Add literature for recent years.
2. What are the prevailing factors that determine the transition from the first to the third evaporation regime in Fig. 4?
3. Why is the temperature on the free surface of the liquid in the central part of the layer lower than at the edges of the layer (Fig. 7)?
4. What heats were taken into account when modeling combustion using carbon dioxide hydrate?
After minor changes as part of the minor revision, the manuscript can be accepted for publication in the journal Fire.
Author Response
|
The introduction has been expanded. Literature and its analysis added. |
|
The transition to the convection regime is associated with the buoyancy of the gas, which depends on the layer diameter and the gas density gradient. The rate of convection significantly increases the rate of evaporation of the fuel. |
|
When burning, the metal walls of the tank quickly heat up. The heat flow from the side walls to the liquid heats the fuel layer near the side walls. The maximum distance from the side walls (colder area) in the center of the layer. |
|
The heat balance takes into account the heat of fuel evaporation, the heat of gas hydrate dissociation, the heat of ice melting and water evaporation from gas hydrate, the heat of radiation, the heat of fuel combustion, and the gas convective heat flux. |
Reviewer 2 Report
Please check the attachment.

Author Response
|
In the Abstract and Introduction, a text has been added on the importance of the subject, as well as the existing groundwork and prospects for further research. Advantages of the new method compared to water spray and other methods were added. |
|
In the Introduction, previous work and its analysis on firefighting with carbon dioxide hydrate were added. The advantages of this method were indicated: In recent years, there has been a noticeable success in the development of technologies for storing and transporting gas hydrates at subzero temperatures, which enables a more efficient (less costly) delivery of inert gas hydrates to the flame region for rapid suppression of combustion. The complicated development of these technologies is also due to the lack of elaborated methods for suppressing combustion, which requires additional experimental and theoretical studies. Combustion suppression with the use of CO2 hydrate has to contend with a variety of related scientific tasks and key factors: convective heat and mass transfer; calculation of free convection, which strongly affects the evaporation rate of fuel; determination of criteria for combustion stability at inert gas hydrate powder entering the combustion region; as well as computation of the dissociation rate of gas hydrates at negative powder temperatures. Earlier, experiments were carried out to extinguish the flame using CO2 hydrate: 1) stopping the flame front, as well as 2) extinguishing a volumetric fire [45, 46]. CO2 hydrate powder demonstrated more effective extinguishing than water spray, snow and ice. A particularly good result was observed for a volumetric fire extinguishing. Small drops of water did not fall into the lower part of the flame, evaporating in its upper part. Solid porous aggregates of gas hydrate fell into the bottom part of the tank until complete dissociation, providing flame suppression (lowering the temperature) over the entire height of the flame.
|
|
The following text has been added to the Abstract: Extinguishing liquid fuels is quite a complicated scientific and technical task. It is often necessary to deal with fire extinction during oil spills and at fuel burning in large containers outdoors and in warehouses. Recently, attention to new extinguishing methods has increased. Advances in the technology of production, storage and transportation of inert gas hydrates enhance the opportunities of using CO2 hydrate for extinguishing liquid fuels. Previous studies have shown a fairly high efficiency of CO2 hydrate (compared to water spray) in the extinction of volumetric fires. |
|
The text has been changed. The effectiveness of the firefighting system were written in more detail. |
|
The introduction corrected. In the Introduction, an analysis of previous studies and the importance of new work on firefighting with carbon dioxide hydrate was added. The advantages of this method and its prospects have been added. |
|
Figure2 corrected. The size of the dots (symbols) is reduced. Designations (a), (d) and (c) are shifted to the center of the figures. Other drawings have also been corrected. |
|
The emission coefficient of fine-grained, porous gas hydrate is quite high (0.94-0.96). The temperature of the free surface of the liquid layer before and after combustion was measured by the infrared camera NEC R500 with the resolution 640 × 512 pixels and an error of 1–2 °C.
|
|
Text corrected. |
|
On Fig. 15b experimental data on the measurement of the maximum flame temperature during the combustion of liquid fuels have been added. A slight underestimation of the temperature is associated with a simplification of the calculation of the combustion kinetics. Ignoring the multistage nature of elementary reactions is justified in this simplified approach, since the main goal of simplified estimates is to determine the amount of gas hydrate required to extinguish a flame. When powder falls on a thermocouple, it is impossible to accurately measure the temperature of the gas. Verification of the general setting of quenching when the gas hydrate powder falls is confirmed by the acceptable agreement between the experiment and the calculation in Fig. 17. In addition, in previous articles, the above technique corresponded to the experimental data on the dissociation rate of various gas hydrates [51, 53, 55, 56, 58], the calculated values of the flame temperature during the combustion of methane hydrate corresponded to the experiment [53, 55, 56]. The data on combustion stability also satisfactorily agreed with the experimental results in the indicated works. |
Reviewer 3 Report
The submitted manuscript studied the process of combustion of a liquid fuel layer (diesel, kerosene, gasoline, separated petroleum and oil) in the presence of CO2 hydrate. For the first time, a method for joint accounting of both the combustion of liquid fuel and the dissociation rate of the falling powder of gas hydrate at negative temperature is proposed. Temperature measurements were performed on the wall and on the free layer surface using thermal imaging technique. This work is interesting, However, to perfect the paper presentation, I think a few revisions should be made as follows:
1. The abstract should be revised to highlight the combustion and extinguishing of liquid fuels.
2. what is the difference of scheme betweeen using the tablets of CO2 hydrate and using CO2 hydrate powder?
3. The authors should address the importance of this work to flame extinguishing process in practice.
Author Response
|
The abstract corrected. |
|
The text at the end of the paper has been added: With an increase in t With an increase in the mass of liquid fuel (with an increase in h), the liquid takes longer to warm up to a quasi-stationary thermal regime. The heating rate of the fuel has a quasi-linear character depending on the layer height (the fuel mass). |
|
In order to show the importance of the work, the following text has been added: 1. In the Introduction: Earlier, experiments were carried out to extinguish the flame using CO2 hydrate: 1) stopping the flame front, as well as 2) extinguishing a volumetric fire [45, 46]. CO2 hydrate powder demonstrated more effective extinguishing than water spray, snow and ice. A particularly good result was observed for a volumetric fire extinguishing. Small drops of water did not fall into the lower part of the flame, evaporating in its upper part. Solid porous aggregates of gas hydrate fell into the bottom part of the tank until complete dissociation, providing flame suppression (lowering the temperature) over the entire height of the flame. 2. At the end of the paper: Thus, the Thus, the jo Thus, the joint accounting of heat and mass transfer and fuel evaporation with free gas convection above the fuel layer, the fuel evaporation rate and the gas hydrate dissociation rate, as well as the calculation of the critical rate of stable combustion, allows applying a simplified methodology to determine the minimum mass of carbon dioxide hydrate required to suppress combustion. A thick layer of liquid fuel, as well as a volumetric fire, is the most difficult to extinguish [46]. As indicated in the Introduction, CO2 hydrate powder shows much greater efficiency compared to a water spray [46]. Large porous aggregates of CO2 hydrate effectively fall deep into the combustion area. However, the growth of the particle diameter, the diameter of the particle aggregate and the height of the falling vapor layer leads to a decrease in the gas hydrate dissociation rate. Optimization of several key factors is necessary for optimal selection of the diameter of aggregates, which is the subject of further research. The use of pressed tablets and CO2 hydrate powder are two extreme cases. Porous powder is best used to extinguish a flame when it is necessary to quickly cover the entire surface of the burning substance and ensure a high rate of carbon dioxide release (from the gas hydrate) for a split second. In this case, the flame height should be relatively low. If the distance to the place of fire is meters-tens of meters and the height of the flame is meters, then the powder with small particles can disintegrate before it reaches the surface of the burning substance. In this case, it is better to use large granules (tablets) of CO2 hydrate, which will begin to disintegrate after falling on the burning surface, bypassing the intense and high flame. The size of the granules and their porosity will depend on the distance to the fire and the height of the flame. To determine the method of CO2 hydrate delivery to the fire site, there is a need in additional experimental and theoretical studies for optimizing the geometric parameters of the gas hydrate and its mass. 3. In the Conclusions: To extinguish a volumetric fire with a high flame height, as well as at a great distance to the fire site (meters-tens of meters), it is efficient to use pressed tablets or granules. In this case, the gas hydrate tablet will not have time to disintegrate until it falls into the lower region of the flame. In some cases of local combustion and low flame height (less than 0.5-1 m), it is possible to use a highly porous powder with CO2 hydrate particles of small diameter (about 1-2 mm or less) to extinguish the flame. |
Reviewer 4 Report
The paper discusses issues related to the combustion process of liquid fuels in the presence of CO2 hydrate powder. The subject of the paper is very interesting. The work has been written in accordance with the requirements of the journal. The literature analysis is extensive. The model and discussion of research results are clear. I believe that the article can be published in its current form.
Author Response
The authors thank the reviewer for a positive attitude to the paper.
Reviewer 5 Report
The paper has two main parts, one is measurements and the other is simplified theoretical, which is called simulation. I think calling it simulation may not be the best practice, because, by simulation, people mainly refer to solutions of detailed equations of the flow and combustion.
While the theoretical approach makes sense, given their primary simplification assumption, some sort of validation from the measurements is required to introduce and include in the paper.
In general, the two parts are very disconnected from each other. some coherence and relevance of the two are needed.
one weakness of the paper is that the reader cannot obtain a general picture after reading the paper and the conclusion section is rather a summary of tasks done in the paper.
The introduction lacks a narrative, even though the literature review is good. I suggest that the authors will rewrite the introduction to have a narrative rather than reporting what has been done in what reference.
line 240: Explain the physical reasoning behind this behavior.
line 357: eq 4 is meant?
line 366-7: any comments on the inter-particle effects?
The quality of English is almost fine, but some editions are required. for example, I noted a sentence that was not complete, even though I could grasp what they authors are talking about (since it is mostly by symbols and equations)
Author Response
|
Thank you for your comment. The word “simulation” was removed from the text of the manuscript, which was replaced by calculation or estimates or simplified estimates. |
|
On Fig. 15b experimental data on the measurement of the maximum flame temperature during the combustion of liquid fuels have been added. A slight underestimation of the temperature is associated with a simplification of the calculation of the combustion kinetics. Ignoring the multistage nature of elementary reactions is justified in this simplified approach, since the main goal of simplified estimates is to determine the amount of gas hydrate required to extinguish a flame. When powder falls on a thermocouple, it is impossible to accurately measure the temperature of the gas. Verification of the general setting of quenching when the gas hydrate powder falls is confirmed by the acceptable agreement between the experiment and the calculation in Fig. 17. In addition, in previous articles, the above technique corresponded to the experimental data on the dissociation rate of various gas hydrates [51, 53, 55, 56, 58], the calculated values of the flame temperature during the combustion of methane hydrate corresponded to the experiment [53, 55, 56]. The data on combustion stability also satisfactorily agreed with the experimental results in the indicated works. |
|
A text has been added to the article linking different parts - combustion and decomposition of gas hydrate, experiment and calculation. Added suggestions for the transition of the experiment to the calculation: To develop a technique for flame extinguishing using CO2 hydrate, it is necessary to solve a number of interrelated tasks, associated with the following areas of research: 1) Heat and mass transfer and evaporation of liquid fuel during combustion, as well as in the presence of inert gas hydrate in the combustion region; 2) Determination of the dissociation rate of gas hydrate at negative powder temperatures; 3) Determination of the boundaries of flame stability when using CO2 hydrate. These problems have not been solved to date, with each being usually investigated separately due to a large number of factors and uncertainty of boundary conditions. However, in order to optimize the extinguishing from an energy point of view (determining the minimum amount of CO2 hydrate powder), it is expedient to solve these problems simultaneously. For the correct solution of the thermal problem, it is important to determine the geometric parameters of the layer when it is necessary to take into account free convection, strongly affecting the fuel evaporation rate and the thermal balance in the combustion region. Therefore, the article examines the influence of the diameter of the liquid fuel layer on the kinetics of evaporation (the value of the exponent linking the layer diameter with the evaporation rate is determined). Combustion kinetics depends on many parameters, including the concentration of fuel and inert gas. The task of determining the fuel concentration in the flame region, as well as in the presence of gas hydrate, is directly related to the determination of the fuel evaporation rate and the dissociation rate of CO2 hydrate at negative powder temperatures. At that, negative temperatures are associated with the need to store the gas hydrate and transfer it to the combustion region without dissociation, until the powder enters the flame. Flame extinguishing is aligned with determining the boundaries of stable combustion, which depend on the heat flux, determination of the combustion temperature, the fuel evaporation rate, and the dissociation rate of gas hydrate. In connection with the above, experimental studies and simplified calculation methods concern all these tasks.
|
|
The text at the end of the manuscript showing the novelty and importance of the results obtained, as well as a comparison with other methods, have been added: Thus, the joint accounting of heat and mass transfer and fuel evaporation with free gas convection above the fuel layer, the fuel evaporation rate and the gas hydrate dissociation rate, as well as the calculation of the critical rate of stable combustion, allows applying a simplified methodology to determine the minimum mass of carbon dioxide hydrate required to suppress combustion. A thick layer of liquid fuel, as well as a volumetric fire, is the most difficult to extinguish [46]. As indicated in the Introduction, CO2 hydrate powder shows much greater efficiency compared to a water spray [46]. Large porous aggregates of CO2 hydrate effectively fall deep into the combustion area. However, the growth of the particle diameter, the diameter of the particle aggregate and the height of the falling vapor layer leads to a decrease in the gas hydrate dissociation rate. Optimization of several key factors is necessary for optimal selection of the diameter of aggregates, which is the subject of further research. Text in the Introduction has been added: Earlier, experiments were carried out to extinguish the flame using CO2 hydrate: 1) stopping the flame front, as well as 2) extinguishing a volumetric fire [45, 46]. CO2 hydrate powder demonstrated more effective extinguishing than water spray, snow and ice. A particularly good result was observed for a volumetric fire extinguishing. Small drops of water did not fall into the lower part of the flame, evaporating in its upper part. Solid porous aggregates of gas hydrate fell into the bottom part of the tank until complete dissociation, providing flame suppression (lowering the temperature) over the entire height of the flame.
The conclusions corrected. |
|
The rationale for the importance of research in the Introduction, the added transitions to different topics, more detailed research objectives, and the added the benefits of using CO2 hydrate powder were added. The literature on the use of CO2 hydrate for flame extinguishing was added. |
|
Thank you for your comment. The inscription to Fig. 6 has been corrected. Additional clarification has been added: With an increase in the mass of liquid fuel (with an increase in h), the liquid takes longer to warm up to a quasi-stationary thermal regime. The heating rate of the fuel has a quasi-linear character depending on the layer height (the fuel mass). |
|
Thanks for pointing out the typo. «Then, Eq.(2) will be simplified to become Eq. (5).» corrected to «Then, Eq.(4) will be simplified to become Eq. (5).» |
|
The sentence: “To simplify modeling, the dissociating CO2 hydrate particles are considered independently of each other.” was deleted. The influence of particles on each other was taken into account in the simulation. For each particle of the layer, the decay rate is calculated. For the entire layer, the balance of mass and heat is considered. Next, a transition is made from one layer of particles to another after a time Δt over the entire thickness h of the powder layer. The powder layer consists of many horizontal layers of particles. |
|
English corrected. |
Round 2
Reviewer 2 Report
The author has made revisions to the manuscript and I believe it can be accepted.
Reviewer 5 Report
The authors have met my expectations